# Conformational and migrational dynamics of slipped-strand DNA three-way junctions containing trinucleotide repeats

Tianyu Hu [1], Michael J. Morten [1] & Steven W. Magennis [1]✉

Expansions of CAG/CTG trinucleotide repeats in DNA are the cause of at least 17 degenerative human disorders, including Huntington's Disease. Repeat instability is thought to occur via the formation of intrastrand hairpins during replication, repair, recombination, and transcription though relatively little is known about their structure and dynamics. We use single-molecule Förster resonance energy transfer to study DNA three-way junctions (3WJs) containing slip-outs composed of CAG or CTG repeats. 3WJs that only have repeats in the slip-out show two-state behavior, which we attribute to conformational flexibility at the 3WJ branchpoint. When the triplet repeats extend into the adjacent duplex, additional dynamics are observed, which we assign to interconversion of positional isomers. We propose a branchpoint migration model that involves conformational rearrangement, strand exchange, and bulge-loop movement. This migration has implications for how repeat slip-outs are processed by the cellular machinery, disease progression, and their development as drug targets.

[1] School of Chemistry, University of Glasgow, Joseph Black Building, University Avenue, Glasgow G12 8QQ, UK. ✉email: steven.magennis@glasgow.ac.uk

Secondary DNA structures that feature extra-helical slip-outs of tandem repeats are key intermediates in repeat expansion diseases (REDs)[1–10]. While isolated hairpins have received most attention, less is known about the structures formed when the hairpin extrudes from the duplex[9]. These branched structures, which are DNA three-way junctions (3WJs), can form during replication, repair, and recombination, and are involved in all current models of repeat expansion. Slip-outs of CTG or CAG repeats are known to form open-loop structures for short repeats[8,11], or intrastrand hairpins (via GC hydrogen bonds) for longer repeats[12–19]. In this work, we investigated the dynamic behavior of these biologically relevant 3WJs.

We used two complementary single-molecule Förster resonance energy transfer (SM-FRET) techniques[20–22] to study 3WJs with slip-outs of between 2 and 30 CTG or CAG repeats. The presence of 3 complementary CAG-CTG triplets in the duplex allows, in principle, four possible Watson-Crick positional isomers to exist, with identical numbers of base pairs; we refer to these as mobile 3WJs. We reveal that these secondary structures undergo two distinct types of dynamics, which has implications for understanding and treating trinucleotide REDs, and their development as drug targets[23,24].

## Results

**Static 3WJs display two-state dynamics.** We first looked at 3WJs that only have repeats in the slip-out, such that the branchpoint is fixed relative to the flanking duplex regions; we refer to these as static 3WJs (Fig. 1 and Supplementary Fig. 1; sequences for all oligos and 3WJs used in this study can be found in Supplementary Tables 1–4). We designed four static 3WJs with slip-outs of 10 repeats of either CAG or CTG that form a stable intrastrand hairpin[12]; these mimic the possible positional isomers that could form for mobile 3WJs having a 10-repeat slip-out (see next section). We used Alexa488 and Cy5 as labels for SM-FRET because we have previously used these dyes for studies of fully complementary 3WJs that contained no tandem repeats[25,26].

Multi-parameter fluorescence detection (MFD) revealed the distribution of FRET states for static 3WJs that were freely diffusing (Fig. 1 and Supplementary Figs. 2–4). The 2D plots of FRET efficiency ($E_{FRET}$) and donor anisotropy ($r_D$) versus donor lifetime ($\tau_{D(A)}$) show that the average FRET value varies according to the expected dye–dye distances for the different structures (Supplementary Table 5), which is good evidence that intrastrand hairpins are formed. For the $(CTG)_{10}$ slipouts, FRET histograms of S1, S2, and S3 could be reliably fitted to three Gaussians, with the peak at zero FRET due to donor-only species. Both of the FRET populations for each static 3WJ lie on the theoretical FRET line which means the FRET states are static or have a very slow exchange rate within the millisecond timescale of single-molecule diffusion through the confocal volume[27]. We previously observed two FRET states for fully complementary

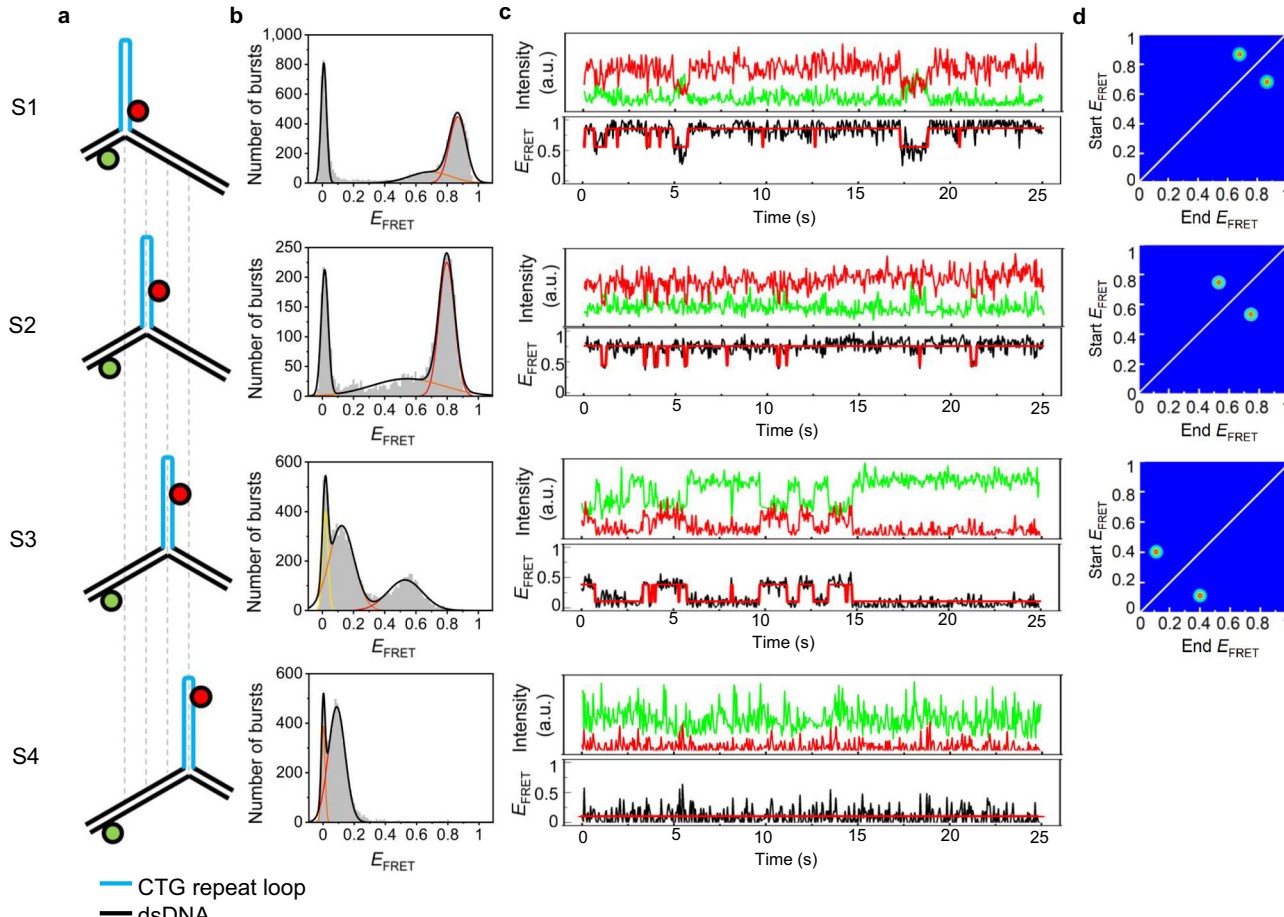

**Fig. 1 Two-state FRET dynamics for static 3WJs. a** Static 3WJs, S1–S4, formed from two DNA strands, labeled with either Alexa488 (green) or Cy5 (red); the $(CTG)_{10}$ slip-out (blue) is fixed relative to the fully complementary dsDNA (black). **b** FRET histograms from MFD of freely diffusing 3WJs fitted to one donor-only state and one (S4) or two (S1–S3) FRET states. **c** TIRF time traces of surface-immobilized 3WJs; Alexa488 (green) and Cy5 (red) intensities, and calculated FRET efficiency ($E_{FRET}$) and idealized trace (red) from Hidden Markov modeling. **d** Transition density plots of TIRF FRET traces. Measured at 20 °C in a buffer containing 1 mM MgCl$_2$.

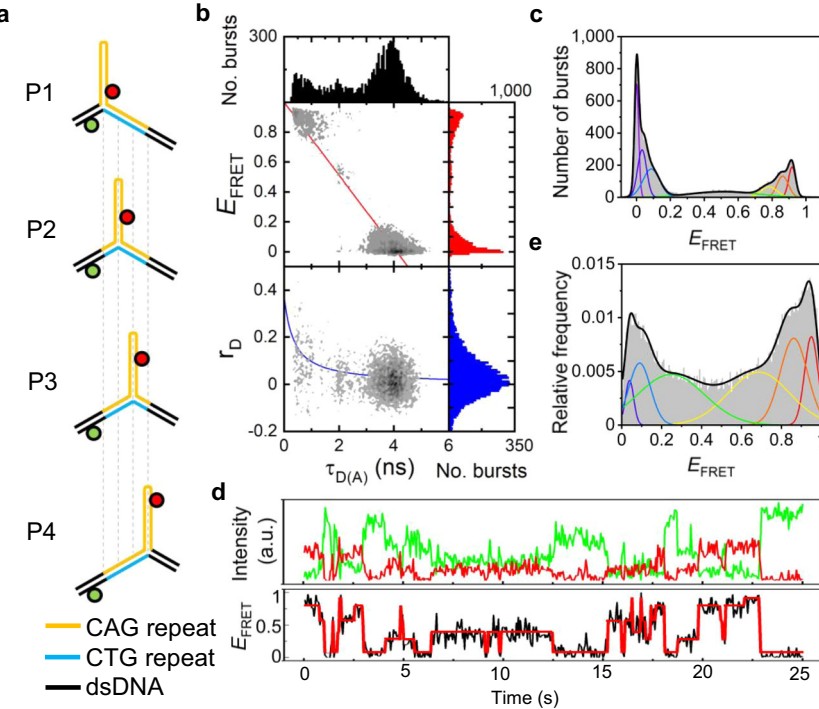

**Fig. 2 Interconversion of positional isomers for mobile 3WJs. a** The presence of three complementary CAG (orange) and CTG (blue) repeats in the duplex and 10 CAG repeats in the slip-out (orange) could result in four positional isomers P1–P4. **b** 2D MFD plot shows FRET efficiency ($E_{FRET}$) or donor anisotropy ($r_D$) vs donor lifetime [$\tau_{D(A)}$] for freely diffusing 3WJs. **c** MFD FRET histogram fits to six FRET states. **d** TIRF time traces of surface-immobilized 3WJs; Alexa488 (green) and Cy5 (red) intensities, and calculated FRET efficiency ($E_{FRET}$). **e** TIRF FRET histogram fitted to six FRET states. Measured at 20 °C in a buffer containing 1 mM $MgCl_2$.

3WJs that lacked any tandem repeats using SM-FRET and NMR[26]; these states were due to conformations with paired and unpaired branchpoints that exchanged at a rate of 15 Hz. On that basis, we assign the two FRET states for the static 3WJs (S1–S3) to different conformers. The FRET distribution of S4 fits to one FRET state, which we attribute to overlapping FRET signals due to the large dye–dye separation in this structure. The four static 3WJs together result in six unique FRET states, since the state of S4 is indistinguishable from one of the S3 states. For the corresponding static 3WJs with $(CAG)_{10}$ slipouts (S-CAG-1 to S-CAG-4), the results were similar, with the exception that two FRET states are observed for all 4 samples; again, only five unique states out of a possible eight states are evident (Supplementary Table 5 and Supplementary Figs. 3 and 4).

Total internal reflection fluorescence (TIRF) microscopy was then used to study the dynamics of immobilized static 3WJs with $(CTG)_{10}$ or $(CAG)_{10}$ slipouts. We observed excellent agreement between TIRF and MFD, with very similar FRET states observed for each 3WJ, showing the surface immobilization did not perturb the 3WJ (Fig. 1, Supplementary Table 5, and Supplementary Figs. 5–7). Notably, where we observed two states, these were interconverting (Fig. 1 and Supplementary Fig. 7).

For the static 3WJs with a $(CTG)_{10}$ slipout, transitions between two FRET states were clearly seen in the time traces of S1, S2, and S3, while a constant FRET signal was observed for S4 (Fig. 1). The rate constants were the same, within error, for S1–S3, with average values (sample standard deviation) of 12.25(0.61) $s^{-1}$ and 3.38 (0.23) $s^{-1}$ for the low-to-high FRET and high-to-low FRET, respectively. Similar results were obtained for the static 3WJs with a $(CAG)_{10}$ slipout; transitions between two FRET states were seen for all four 3WJs, with average values (sample standard deviation) of 11.74(0.93) $s^{-1}$ and 4.35(0.73) $s^{-1}$ for the low-to-high FRET and high-to-low FRET, respectively. These rate constants are similar to

the 15 Hz exchange rate observed by NMR for fully complementary 3WJs[26]. These time traces were used to produce transition density plots (TDPs) of state-to-state connectivity, which clearly show the two-state transitions of S1, S2, and S3 and that all transitions are reversible (Fig. 1). A variant of S3 with an additional 10 repeats ($CTG_{20}$) in the slip-out had the same FRET distribution as S3 confirming hairpin formation (Supplementary Fig. 8 and Supplementary Table 6). The rate constants for this variant were 11.71 (0.58) $s^{-1}$ and 2.69 (0.17) $s^{-1}$ for the low-to-high FRET and high-to-low FRET, respectively; thus, the conformational dynamics are very similar for slip-outs with 10 and 20 repeats. We also designed a version of S3 with fully complementary base pairs in the hairpin. The MFD data and TIRF are qualitatively similar with major differences only in FRET efficiencies, presumably due to the increased pairing compacting the length of the slip-out hairpin (Supplementary Fig. 9 and Supplementary Table 6). The rate constants of 11.43(0.55) $s^{-1}$ and 2.38(0.43) $s^{-1}$ for the low-to-high FRET and high-to-low FRET, respectively, are in good agreement with those for S1–S4, which is further evidence of hairpin formation in the slip-out structures. SM-FRET was not strongly dependent on the presence of $Mg^{2+}$ ions, though additional FRET states did appear at high concentrations, presumably due to other ion-induced conformations (Supplementary Figs. 10 and 11).

**Interconversion of positional isomers.** Mobile 3WJs were designed in which CTG or CAG repeats extended from the slip-out into the fully complementary duplex by 3 repeats; the slip-outs contained 2–30 repeats (Fig. 2a and Supplementary Fig. 12). We chose even numbers of repeats since these are known to form stable hairpins with a tetranucleotide loop[28]. Each mobile 3WJ can, in principle, exist as one of four positional isomers (P1–P4) via shifting of the location of the slip-out relative to the two fully complementary dsDNA arms (Fig. 2a). For a mobile 3WJ with

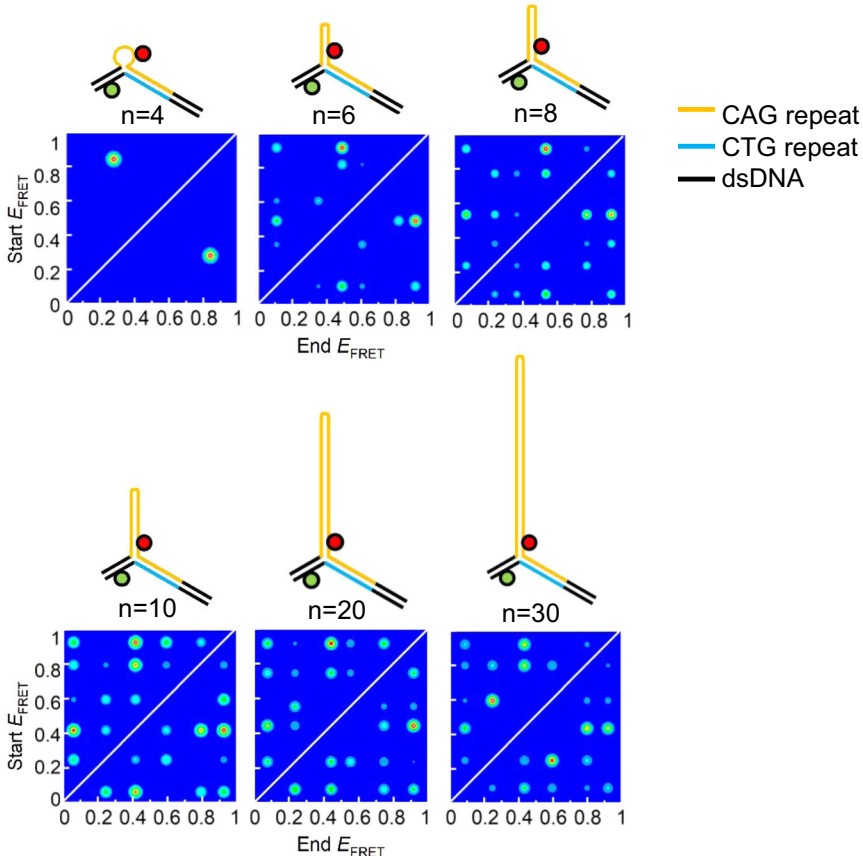

**Fig. 3 Dependence of dynamics on repeat length.** Transition density plots from TIRF time traces of 3WJs with three complementary CAG (orange) and CTG (blue) repeats in the duplex and $(CAG)_n$ slip-outs ($n = 4$, 6, 8, 10, 20, and 30). 3WJs with slip-outs of $(CAG)_4$ have two FRET states, while all other 3WJs are fitted to six FRET states. Similar plots are found for 3WJs with $(CTG)_n$ slip-outs (Supplementary Fig. 20). Measured at 20 °C in a buffer containing 1 mM $MgCl_2$.

$(CAG)_{10}$ or $(CTG)_{10}$ slip-outs, these positional isomers are identical to the static 3WJs (S1–S4), with the exception of 9 base pairs at the branchpoint (Fig. 1a). The MFD data for $(CAG)_{10}$ slip-outs (Fig. 2b, c) are representative of those for mobile 3WJs, displaying multiple FRET states with a wide range of FRET efficiencies (see Supplementary Figs. 13–16 for other mobile 3WJs). As the number of repeats increases, the number of FRET states required for satisfactory fitting of MFD data increases from one ($n = 2$) and two ($n = 4$), to a maximum of six FRET states for 6 or higher repeats. For $n = 8$, 10, 20, or 30, the six FRET states are all of very similar value and match the six FRET states found for the static 3WJs (Supplementary Table 5). This strongly suggests that the mobile 3WJs with more than 6 repeats in the slip-out can adopt any of the four possible positional isomers (Fig. 2a). The fewer number of states observed for $n = 2$ and 4 is attributed to the formation of bulged loop isomers[11,29,30], rather than an intrastrand hairpin, which results in overlap of FRET states. This overlap could be due to the more compact structure and/or the rapid interconversion of positional isomers. As with the static 3WJs, high concentration of $Mg^{2+}$ ions resulted in small changes to the histograms, while experiments conducted at 37 °C were similar to those at 20 °C (Supplementary Fig. 17).

With TIRF, we found the same number of FRET states for mobile 3WJs as observed by MFD (Fig. 2, Supplementary Table 5, and Supplementary Figs. 18 and 19). Crucially, in all cases where more than one FRET state was observed ($n > 2$) almost all possible transitions between FRET states were observed (Fig. 3, Supplementary Fig. 20, and Supplementary Tables 7 and 8). Thus, in addition to the conformational change observed for static

3WJs, the mobile 3WJs can also interconvert between positional isomers. Furthermore, the 3WJs can make jumps to positional isomers that are at least 2 positions away (e.g., P1 to P3). Since P3 has a FRET state that matches that of P4, we cannot rule out larger rearrangements (e.g., P1 to P4). The heat maps for the mobile 3WJs (Fig. 3, Supplementary Fig. 20, and Supplementary Tables 7 and 8) are generally highly symmetrical about the diagonal showing these transitions are reversible. It can also be seen from these heat maps that the transitions are not equally weighted but vary depending on sequence and repeat length. The only exception to the dynamic behavior of these mobile 3WJs was for the $(CTG)_{10}$ slip-out, which displayed a single low FRET state corresponding to isomer P4. Since the Cy5 dye is near the loop in P4, and since Cy dyes are known to stack on DNA[31,32], we attribute this to a specific dye interaction for Cy5. In support of this, we looked at two different control samples, which both displayed the rich dynamics that we observed with all of the other mobile 3WJs. In one case, we switched the Cy5 acceptor dye for Atto647N, which resulted in six interconverting FRET states being observed (Supplementary Fig. 21). In a second control, we moved the attachment position of the Cy5, such that it was never close to the loop in any of the positional isomers; this 3WJ resulted in FRET states and dynamics that were similar to those of the Atto647N-labeled 3WJ (Supplementary Fig. 22).

## Discussion
DNA secondary structure formation underlies all proposed mechanisms for triplet repeat expansion, yet there is relatively less

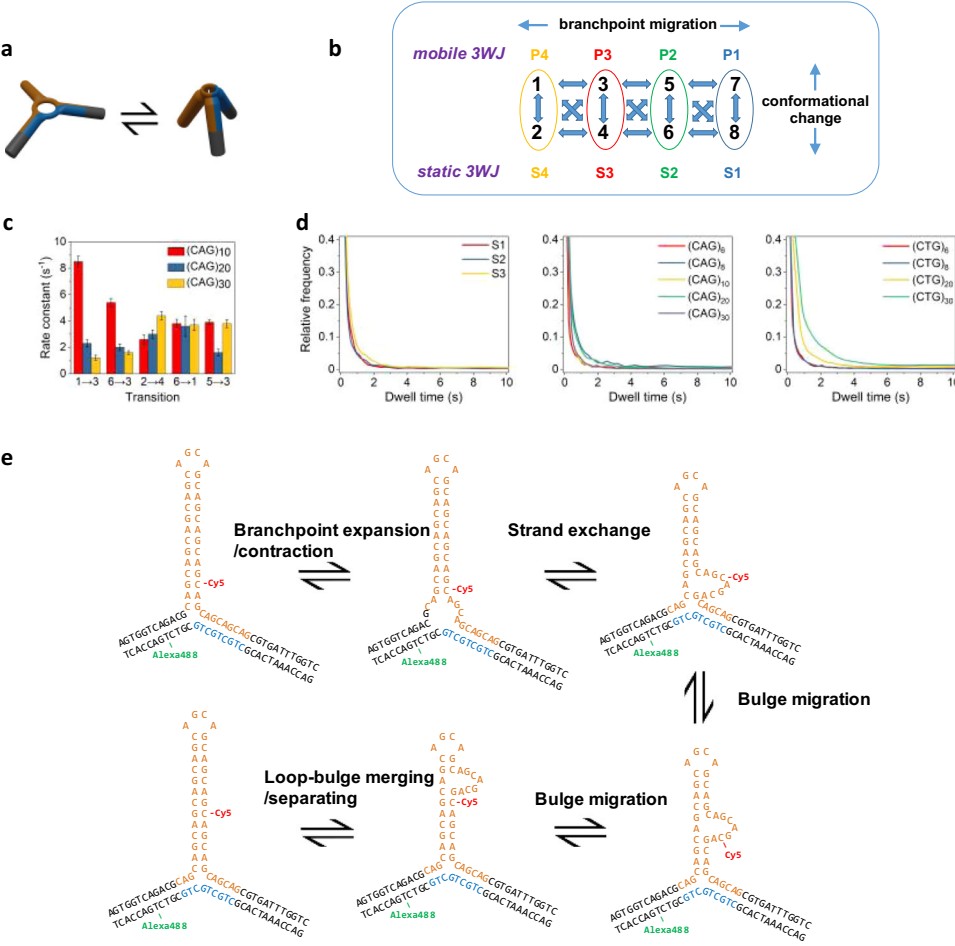

**Fig. 4 Conformational and branch migration dynamics in slip-out 3WJs. a** 3WJs with hairpin slip-outs can undergo conformational change, attributed to branchpoint rearrangement (e.g., expansion/contraction). **b** Kinetic scheme showing the eight possible FRET states as a result of the two conformations for each positional isomer. **c** Selected rate constants for mobile 3WJs with 10, 20, or 30 repeats in the slip-out. Error bars represent the standard error from non-linear curve fitting (see Methods for details). **d** Cumulative dwell times histogram for static 3WJs (left), and mobile 3WJs with CAG (middle) or CTG (right) slip-outs. **e** Model of branch migration in 3WJs with trinucleotide repeats in a slip-out and in the duplex. Measurements made at 20 °C in a buffer containing 1 mM MgCl₂.

information about their structure and dynamics. There is some debate in the literature regarding the minimum number of slip-out repeats required to form intrastrand hairpins possibly reflecting the sensitivity and bias of different techniques, and also differing sequence contexts. However, it is generally accepted that between 6 and 15 repeats are sufficient, with CTG being more stable than CAG[12–18].

It was shown previously using SM-FRET and NMR that fully complementary 3WJs lacking triplet repeats could adopt two conformations that were attributed to open and closed states at the branchpoint[26]. NMR was also used to show that multiple conformations, including unpaired bases, can exist at the branchpoint of static 3WJs that contain triplets, and that the branchpoint conformation could affect DNA repair[9]. Furthermore, two interconverting conformations in static 3WJs containing 13 CAG repeats were detected by SM-FRET and were postulated to regulate mismatch repair[33]. Therefore, we attribute the FRET dynamics observed for static 3WJs to changes in global structure that is due to local rearrangement at the branchpoint (Fig. 4a); this behavior also occurs in the mobile 3WJs.

For mobile 3WJs, we have shown that those with 6 or more repeats of either CAG or CTG form structures with FRET states that match those of the corresponding static structures (Fig. 4b). Mobile 3WJs with slip-outs composed of CAG or CTG repeats

undergo two types of dynamics: (1) two-state conformational dynamics that is attributed to local branchpoint rearrangement and (2) interconversion of positional isomers, whereby the branchpoint shifts by one or more repeat units (Fig. 4b). The dynamic behavior was very similar for CAG and CTG slip-outs. The similar values of the FRET efficiency for each state shows that the global structure around the branchpoint is similar (at least for 6–30 repeats).

Although eight FRET states are predicted, we were only able to reliably fit up to six FRET states, which we attribute to overlapping FRET states and the resolution of the method. For six FRET states there are 30 unique transitions possible. In general, most transitions were observed and all were reversible (Fig. 3, Supplementary Fig. 20 and Supplementary Tables 7 and 8), Dwell-time distributions of many of the most frequent transitions could be fitted to single-exponential decays (Fig. 4c). An overlap in FRET states would mean that multi-exponential decays were also possible; however, in this work we have assumed that Markovian kinetics apply throughout. Although there were some trends as a function of repeat length, there was not a clear pattern. Given the similar sequence contexts at the branchpoint for the four positional isomers, we do not expect a strong bias for particular isomers as has been noted for fully complementary 3WJs[34]. More was revealed by combining data from all

transitions to produce cumulative dwell-time distributions (Fig. 4d), Conformational change and positional interconversion occur on similar timescales for mobile 3WJs up to $n = 10$, and match the dwell times for the static 3WJs (Fig. 4d). However, for 20 and 30 repeats in the slip-out there is a marked increase in the dwell times for CTG versus those of the other 3WJs (Fig. 4d). This difference agrees with the higher stability for CTG intra-strand hairpins compared with those of CAG[18].

SM-FRET has been used previously to study the dynamics of isolated hairpins containing CAG or CTG repeats. Hairpins with 6–12 repeats of CAG or CTG separated by a polyA or polyT linker showed hairpin opening and closing transitions between the fully paired conformations, with CTG structures being more stable[35]. While short slip-outs could shift by the opening and closing of only a few base pairs[11], this is increasingly unlikely as the number of repeats increase. Based on SM-FRET studies of hairpins with CTG repeats[36] and TGGAA repeats[37], a bulge translocation model has been proposed. This is supported by SM-FRET studies of isolated CAG hairpins which showed that even numbers of repeats were stable while odd numbers of repeats were frustrated and could shift in register between blunt-ended hairpins or those with a CAG overhang[28].

Based on the prior work discussed above and the evidence presented herein, we propose a model of reversible branch migration in mobile 3WJs with trinucleotide repeats (Fig. 4e). This process is initiated by unpairing at the branchpoint, which is known to be flexible[9,25,26], and is followed by strand swapping and then translocation of a bulge; the end result is a 3WJ that has shifted the location of the slip-out by exactly one trinucleotide (Fig. 4e). Although the conformational change may be decoupled from migration, there is also the intriguing possibility that the conformation switches the branch migration on or off. Such a mechanism is reminiscent of the gating of 4-stranded Holliday junction branch migration via ion-induced conformational change[38].

The importance of secondary structure and the sequence context of slip-outs on processes such as DNA repair is well established[1,3]. The preference for particular positional isomers has been observed previously[8], while repair has been shown to depend on composition and proximity to nicks[19]. The ability of slip-outs to migrate to different locations could determine how they are processed by the cellular machinery, and how they respond to damage[39,40], such as oxidation of guanine[12,41]. Our data reveal that migration of CTG and CAG slip-outs can occur and that not all positional isomers are equally favored.

The largest slip-outs studied here contain 30 repeats which is close to the threshold between the healthy and symptomatic ranges for a number of REDs involving CAG or CTG repeats[1]. The clear slowing down of dynamics of the CTG slip-out with 20 or 30 repeats suggests a possible role of branch migration in disease progression. Similar dynamic behavior may also occur for slip-outs of other tandem repeats. Furthermore, trinucleotide repeats are now showing promise as drug targets[23,24]. Knowledge of the underlying structure and dynamics of repeat structures could be crucial for developing therapies that can halt or reverse repeat instability.

## Methods

**Preparation of three-way junctions.** Oligonucleotides were synthesized and labeled by Purimex GmbH (Grebenstein, Germany) and IBA GmbH (Göttingen, Germany). The NHS-esters of Alexa488 (5′/6′ mixed isomer, Invitrogen), Cy5 (GE-Healthcare), or Atto647N were attached via a 5-C6-aminoallyl-dC or 5-C6-ami-noallyl-dT. The sequences of oligonucleotides are shown in the Supplementary Tables 1–4.

Annealing of samples for branched DNA was carried out in buffer (20 mM Tris (Sigma-Aldrich) and 50 mM NaCl (Fluka), pH 7.5) with the ratio of donor strand to acceptor strands at 1:2. Samples were heated to 90 °C in a water bath and left to cool

down slowly overnight. For MFD measurement, all samples were diluted into buffer containing 20 mM Tris, 15 mM NaCl, and 1 mM ascorbic acid (Fluka) at pH 7.5. Before the addition of sample, the buffer was stirred and cleaned with activated charcoal to remove fluorescent impurities. For TIRF measurement, the buffer contained 20 mM Tris-HCl, 10 mM NaCl (pH 7.8) with 6% glucose (w/w), 2 mg/mL glucose oxidase, 0.08 mg/mL glucose catalase, and 1 mM Trolox to reduce the rate of blinking and photobleaching of the dyes. The measurement buffer contained various concentrations of MgCl2 (Fluka) from 0 mM to 100 mM. All measurements were recorded at 20 ± 1 °C or 37 ± 1 °C.

**Objective-type total internal reflection fluorescence (TIRF) microscopy.** Single-molecule measurements were performed using total internal reflection fluorescence (TIRF) microscopy on a custom-built inverted microscope using objective-type TIRF[42]. Microfluidic flow cells containing immobilized DNA were mounted on an inverted microscope (IX71, Olympus). Cleaned glass coverslips were coated with PEG and PEG-biotin, followed by the addition of neutravidin, and then the biotinylated DNA was added at a concentration of ~10 pM and immobilized via the biotin–neutravidin interaction. This gave ~200 visible spots in the donor channel. An enzymatic oxygen scavenging system (glucose oxidase/catalase system) and Trolox were used to suppress the rate of photobleaching and blinking, respectively. Excitation light (488 nm Stradus (Laser 2000, UK) diode laser) was provided via the evanescent wave of a totally internally reflected beam. Fluorescence emission was collected using a 100 × 1.49 numerical aperture oil immersion objective lens (Olympus) and separated from scattered excitation via a 525/50 nm bandpass filter and a 500 nm dichroic mirror (Chroma Technology Corp.). Fluorescence was subsequently collimated and separated by donor and acceptor emission wavelengths with an emission splitting system (DV2 Multi-channel Imaging System, Photometrics) and EMCCD camera (Evolve, Photo-metrics). TIF movies were recorded by ImagePro-Plus v7.0 software using an exposure time of 50 ms. The fluorescence intensity of the fluorophores was obtained using TwoTone (v3.1)[43]. Anti-correlated donor and acceptor intensities are expected for a FRET-related process and were analysed using hidden Markov modelling (HMM, HaMMy v4.0)[44]. Several hundred single-molecule traces were typically analysed. Traces were stitched for analysis; analysis of unstitched traces gave similar results. The HMM program gave start FRET, end FRET, and intensity ratios of each transition, which were imported into TDP (v3.0) to give a histogram that was plotted in Origin v2020b; the minimum transition number in the TDPs was set to 10, with the exception of CTG10-Atto (Supplementary Fig. 21), where it was 2 due to the smaller number of traces collected. The values of FRET efficiency were unrestrained for all fits with evenly distributed starting values. The number of FRET states for HMM was chosen in order to ensure good agreement with the TIRF and MFD histograms (2 states for static 3WJs and mobile 3WJs with 4 repeat slip-outs, and 6 states for all other 3WJs). Dwell-time distributions were analysed using non-linear curve fitting (Levenberg-Marquardt) in Origin v2020b; the error bars in Fig. 4c represent the standard error scaled by the square root of the reduced $\chi^2$.

**Multi-parameter fluorescence detection.** MFD is based around the burstwise detection of fluorescence as single molecules diffuse through the focus of a confocal microscope. Photon counting detection by multiple detectors allows simultaneous detection of the color, lifetime, polarization, and intensity of fluorescence for each molecule[21]. MFD measurements were performed using a home-built system[45].

The fluorescent donor molecule (Alexa488) was excited by a linearly polarized laser (483 nm, 40 MHz, ~60 ps FWHM; Picoquant, Germany). The laser light was focused into the dilute solution of labeled molecules by a water immersion objective (UPLAPO 60x, NA = 1.2, Olympus, UK). The diffusion time for Rhodamine 110 was ~200 μs (measured using a ALV-7002 hardware correlator and ALV-Correlator software v3.0, ALV GmbH, Germany). The average laser power at the sample was ~170 μW. Each molecule generates a burst of fluorescence photons as it diffuses the detection volume. This photon train is divided into parallel and perpendicular components via a polarizing beamsplitter (Linos, Germany) and then into wavelength ranges above and below 595 nm by using a dichroic beamsplitter (z595 DCXR, Chroma, USA). Additionally, red (HQ 710/130 nm, AHF Analysentechnik, Germany) and green (HQ 525/50 nm, Chroma, USA) bandpass filters in front of the detectors make sure that only fluorescence photons from the acceptor and donor molecules are registered. Detection is performed using four avalanche photodiodes (SPCM-AQRH-14, Perkin Elmer, USA for the red and PDM50C, MPD, Italy for the green). The signals from all detectors are routed, via delay lines, to two synchronized time-correlated single photon-counting boards (SPC 132 cards with SPCM v8.82 software, Becker and Hickl, Germany) connected to a PC. Bursts of fluorescence photons are distinguished from the background of 1–2 kHz by applying threshold intensity criteria (0.1 ms interphoton time and minimum 100 photons per burst). Bursts during which bleaching of the acceptor occurs are excluded from further analysis by applying a criterion in regard to the difference in macroscopic times, $|T_G - T_R| < 0.55$ ms, where $T_G$ and $T_R$ are the average macroscopic times in which all photons have been detected in the green and red channels, respectively, during one burst. Measurements were recorded in Nunc Lab-Tek II chambered cover glasses. Data analysis was carried out using software written by the group of Prof. Claus Seidel at the Heinrich Heine Universität, Düsseldorf (Elke 2002-2003 for burst analysis, Margarita v8.5 for data

visualisation and Kristine v8.1 for FCS) to extract the total signal counts in the green and red emission channels ($S_G$ and $S_R$) and the average counts rates from the parallel and perpendicular green detectors, and the donor lifetime ($\tau_{D(A)}$).

**Reporting summary**. Further information on research design is available in the Nature Research Reporting Summary linked to this article.

## Data availability

The data that support the findings of this study (MFD data for samples and controls, and TIRF time traces) are available from the University of Glasgow data repository Enlighten (https://doi.org/10.5525/gla.researchdata.1086). Source data are provided with this paper.

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

## Acknowledgements

T.H. was supported by the University of Glasgow and the China Scholarship Council. M.J.M. was supported by the Engineering and Physical Sciences Research Council (grant EP/L027003/1).

## Author contributions

All authors designed and planned the experiments. M.J.M. performed preliminary SM-FRET experiments with static and mobile 3WJs, and validated the approach. T.H. prepared the samples and performed the experiments described in this manuscript. S.W.M. conceived the project. All authors analysed the data and prepared the manuscript.

## Competing interests

The authors declare no competing interests.
