## [Peer Review File · Nature Communications]

REVIEWER COMMENTS

Reviewer #1 (Remarks to the Author):

This paper uses single molecule FRET to examine conformational dynamics of DNA 3-way junctions (3WJ) where two arms are matched DNA and the third arm contains a triplet-repeat sequence hairpin. By including a few extra complementary triplet repeats beside the hairpin, the authors can force a few repeats to anneal in the adjacent duplex and observe dynamic stepping of the slipped-out hairpin by one triplet repeat unit, or effectively the 3WJ migrating along the DNA.

The key observations are:

-- For 3WJ lacking the extra complementary repeats, so the junction remains non-migratory, two interconverting states are observed in the FRET between the trinucleotide hairpin and another DNA arm (Fig 1). This behavior generally agrees with the authors previous observations of 3WJ formed from completely matched DNA (not containing trinucleotide repeats) (ref. #26) where the two FRET levels arise from two different conformations with or without basepairing of a few pairs in the junction region. They test four 3WJs designed to mimic the possible positions of the slip-out in the migratory constructs, thus "calibrating" the expected FRET levels in the migratory substrates. Due to the lack of sensitivity of FRET at long distances, these 8 possible states only lead to 6 distinguishable FRET levels.

-- For the migratory 3WJ containing 3 extra CTGs, interconversions among states are observed for repeat numbers above 4. The 6 expected FRET levels from the "calibration" static samples are observed. (Fig 2 and 3)

-- For the migratory 3WJ, the kinetics of the transitions for different lengths of slip-out trinucleotide suggest that all states are visited and most transitions are reversible. (Figure 3, 4 and extended figures)

Based on these observations and previous literature, the authors propose a model for the migratory 3WJs involving reversible branch migration with transient unpairing at the branchpoint followed by strand-swapping and migration of the hairpin (Figure 4).

Overall, the experiments appear to be expertly done and the findings are confirmed by obtaining consistent results when using two different single molecule fluorescence approaches: MFD FRET on freely diffusing molecules and TIRF FRET on immobilized molecules.

The biophysical characterization of the dynamics of these structures is interesting. In principle, the behavior of structures formed from trinucleotide repeat DNA is important. Research activity focused on diseases related to triplet repeat expansion is extensive and growing, so there is likely a large interested group of readers. The biggest weakness of the paper is that the direct biological impacts on understanding these diseases, for example on the speculated drug target development, are limited. Nevertheless, directly observing the migrating 3WJ containing trinucleotide repeats is quite interesting, and I generally support publication. I have a few points the authors may wish to consider.

1) Methodological detail - The method of fluorescent dye attachment to the DNA is not clear to me. All the drawings and oligo sequences suggest the dyes are attached to cytosines (C). The methods say "The NHS-esters of Alexa488 (5'/6' mixed isomer, Invitrogen) or Cy5 (GE-Healthcare), or Atto647N were attached via a 5-C 6'-aminoallyl-deoxythymidine." Isn't that a T base? Am I confused or could the authors clarify this point?

2) Methodological detail – The surface passivation and immobilization scheme for the TIRF experiments is not given. Do the authors use the PEG/PEG-biotin surface, BSA-biotin, neutravidin directly coated on a glass slide, or something else?

3) My biggest concern relates to the dramatically different behavior of the Cy5 labeled (CTG)₁₀ (N=10) slipout 3WJ. It is fascinating that this one sample was completely stabilized in the P4 state and that changing to Atto647N restored behavior in line with all the other samples. The authors' explanation is "Since the Cy5 dye is near the loop in P4, and since Cy dyes are known to stack on DNA (ref. 31,32) we attribute this to a specific dye interaction for Cy5 that is absent when Atto647N is the acceptor." This explanation seems too simple. Looking at the diagrams in Supplementary figure 5 (translating CAG to CTG), it appears that the relative position of the dye and the loop is the same in the P3 state for N=8 3WJ, in the P2 state for N=6 3WJ and in the P1 state for N=4 3WJ as it is in the P4 state for the N=10 3WJ. These other 3 states for the other CTG 3WJs are not stabilized by such positioning (Supplementary Fig. 7 and 9). Why does the Cy5 not similarly stabilize these other states as it does for the P4 state for N=10? There must be a more complex explanation than the one given by the authors in the text.

Regardless of the detailed explanation, the large difference in dynamics for (CTG)₁₀ (N=10) 3WJ labeled with Cy5 compared to Atto647N confirms that the dye can impact the transitions among states in these 3WJs. This effect reduces the overall confidence in the quantitative kinetics reported for all the samples in the manuscript. The confidence in the kinetic parameters could be increased by some other control experiments, for example observing that the kinetics for some other 3WJs are consistent when labeling with Cy5 compared to labeling with Atto647N, or by observing consistent results in a single 3WJ that is labeled with Cy5 at different positions. Without such studies, there is little known about the influence of dye-DNA interaction in other experiments and the accuracy of the kinetic results is questionable.

I commend the authors for identifying this issue with Cy5 in the (CTG)₁₀ 3WJ, finding a solution by changing to Atto647N, and reporting the issue in the paper. This effect of the dye impacting the balance of states of these 3WJs does not reduce the publishability of the study, but I do suggest it warrants more extensive discussion in the text and also reduces overall confidence in the quantitative analysis of the kinetics.

A few other minor points the authors may consider are:

a) The authors may want to be carefully examine their language at several points in the text. For example, in the first sentence of the discussion, they say "DNA secondary structure formation underlies all proposed mechanisms for triplet repeat diseases, ..." I agree that it underlies all proposed mechanisms for DNA triplet repeat expansion, but there are various models of how the diseases arise. Some do not strongly implicate the trinucleotide structure directly in the disease mechanism generating the pathology killing cells. For example, a Cynthia McMurray model of Huntington's [PMID: 30930170] suggests that the induction of double strand breaks due to metabolic processes modified by mismatch repair activity in response to triplet repeat presence is the actual cause of the pathology in Huntington's disease. I am not insisting this particular paper be cited, but rather just suggesting caution in the precise language used.

b) In addition to the issue of dye-DNA interactions altering the energy balance of the states (point 3 above), I am not completely satisfied with how the kinetics studies are done in the paper. No real conclusions depend on the precise kinetics results, so this is just a comment for the authors as they consider possible revisions. The HMM program used to extract kinetics from the TIRF traces seems to not do a great job. From the few time traces in the paper, I see the HMM analysis finding a lot of transitions that could be noise fluctuations (For example Fig 2d, Extended fig 3a, extended fig 8c, etc.) because HMM indicates transitions where there is not anti-correlation in the donor and acceptor intensities. I know HMM is a standard program used widely in the field, but it does not seem to be doing a particularly good job with this data. For that reason, the absolute values of the kinetic parameters extracted do not seem highly confident, but the authors don't make major conclusions based on these kinetics, so maybe it is not such a big deal.

c) related - the transition density plots for the cases without clear transitions seem non-intuitive to me (For example, Figure 3 n=2 case or extended figure 7a and e). What is a transition density plot without obvious transitions in the time traces?

d) related - the selected traces seem to show inconsistent kinetics. This may just be a result of the specific few traces selected. For example, in figure 1 for the non-migratory 3WJ, S2 has very infrequent transitions whereas S3 has more frequent transitions, but looking at the dwell time plots in Figure 4d, and the kinetic rates in the tables, the differences in S2 and S3 look almost negligible.

e) related - Intriguingly, in figure 2d, that trace seems to have some memory effect. From 0 to 20 seconds, it is mostly high FRET with transitions frequently to low FRET, whereas after 20 seconds, it is very low FRET with only a few transitions to higher FRET. The kinetics seem extremely rich in this system and are not captured well in the analysis. Regardless, very few conclusions depend on the kinetics, so maybe this is not a significant criticism.

f) The loss of the mid FRET populations in zero magnesium chloride (extended figure 6 a and e) made me wonder what the TIRF trace of some of these molecules would look like in the absence of magnesium. Not a major issue, but I would have welcomed inclusion of magnesium-free intensity time traces as additional examples.

g) Why do the authors choose CTG slipouts for the static 3WJ experiment and CAG slipouts for the mobile 3WJ experiments for the main figures in the paper? It might not matter, but it would provide more rigor in mapping the FRET states between the experiments instead of putting the mobile CTG slipouts in the supplement or alternatively use CAG static slipouts in the main text. It is probably ok, but given the difference in Cy5 labeled (CAG)10 and (CTG)10 mobile 3WJs, there might be surprises.

h) Possible typo – in captions of extended figure 2e and extended figure 3e, do they mean TIRF FRET instead of MFD FRET?

i) Possible typo– in captions for Supplementary figure 2d, do they mean S4? S3 is repeated twice in the caption.

j) Occasionally the color scheme is not consistent within the figures of the paper. For example, in supplementary figure 4, sometimes purple is the highest FRET state, but sometimes it is not.

Reviewer #2 (Remarks to the Author):

The manuscript entitled “Conformational and migrational dynamics of slipped-strand DNA three-way junctions containing trinucleotide repeats” by Magennis et al. reported the dynamic branchpoint migration and conformational changes at the 3WJs involving the trinucleotide repeat CAG by using single molecular FRET. The dynamic behavior of static 3WJs was obtained with (CTG)10 slip-out. With inserting three CAG repeat against the CTG template, the dynamic behavior of mobile 3WJs was measured. With these SM-FRET data in hand, authors eventually proposed the model of branch migration in 3WJs with trinucleotide in a slip-out and the duplex.

This manuscript focused on the dynamic conformational and migrational behavior of 3WJs possibly produced in the trinucleotide repeats such as CAG and CTG repeats. Measurements of the single-molecule FRET signals of the Cy5-Alexa488 pair and precise analysis of the dynamics revealed two states or six FRET states for static and mobile 3WJs, respectively. The SM-FRET system authors used have been developed previously and well worked for the study of 3WJs with fully complementary duplex with the exchange rate of about 15 Hz in a paper published. Thus, this reviewer assumed that most of the technical issues of the SM-FRET system would be already addressed and solved. Therefore, focused on the 3WJs involved in the trinucleotide repeats.

1) For SM-FRET measurements, it is inevitable of labeling the target, oligonucleotides, in these studies. For the fully complementary duplex, I assume that there would be no interaction of fluorophore and nucleotide bases, but in the hairpin region of CAG trinucleotide repeats, there is an Adenine-Adenine mismatched base pair next to the Cy5-labeled cytosine. I would like to see the comments from the authors if there were any possibilities of interaction between the Cy5 chromophore and the A-A mismatch. If yes, how the interaction affected the bulge migration through the Cy5 labeled cytosine.

2) In figure 3, the authors investigated the dependence of dynamics on repeat length by changing the repeat number from 2 ($n = 2$) to 30 ($n = 30$) with being the position of the labeling cytosine fixed for four CAG repeat away from the end of the repeat. (leaving 3 CAG repeat for hybridization with 3 CTG repeats). Thus, once the branchpoint migration passes the labeled cytosine, the distance between two fluorophores would likely not be much different regardless of the repeat length. While six FRET states were observed for the 3WJs with CAG repeat longer than 6, but plots showed different transition intensities for those showing six FRET stage. I do not see the discussion on these observations.

3) In the last two paragraphs in the discussion, the authors discussed the relevancy of their observations and the possible repair processes. It is interesting, but there is a crucial difference between the possible slip-out structures formed in trinucleotide repeats in the genome and their experimental models, and that is the presence of slip-out in the opposite strand. In the expanded CAG/CTG repeat, both CAG and CTG strands may form the CAG- and CTG-hairpin structures (slip-out) after dissociation of CAG/CTG duplex for the replication and transcription. The dynamic behavior of the CAG slip-out is likely heavily affected by the dynamic behavior of the CTG slip-out in the opposite strand. In the current experimental model, there is no fluctuation conceivable in the CTG strand. Therefore, their discussion on the relevancy between their observations and the repair process would be much carefully discussed.

4) The recent publication (ref. 10) by Nakamori et al. described the CAG-binding small molecules induced the repeat contraction in vivo. It would be particularly interesting how CAG-binding small molecule affects the slip-out dynamics.

**Response to reviewers' comments for Nature Communications manuscript
NCOMMS-20-33279-T**

“Conformational and migrational dynamics of slipped-strand DNA three-way junctions containing trinucleotide repeats” by Tianyu Hu, Michael J. Morten, and Steven W. Magennis

REVIEWER #1

This reviewer was very positive about our manuscript:

“Overall, the experiments appear to be expertly done and the findings are confirmed by obtaining consistent results when using two different single molecule fluorescence approaches.....The biophysical characterization of the dynamics of these structures is interesting. In principle, the behavior of structures formed from trinucleotide repeat DNA is important. Research activity focused on diseases related to triplet repeat expansion is extensive and growing, so there is likely a large interested group of readers.

“...directly observing the migrating 3WJ containing trinucleotide repeats is quite interesting, and I generally support publication.”

This reviewer had **“a few points the authors may wish to consider.”**

We thank the reviewer for their very constructive review and we have considered all the points carefully and amended the manuscript accordingly, as detailed below.

Comment 1) Methodological detail - The method of fluorescent dye attachment to the DNA is not clear to me. All the drawings and oligo sequences suggest the dyes are attached to cytosines (C). The methods say “The NHS-esters of Alexa488 (5'/6' mixed isomer, Invitrogen) or Cy5 (GE-Healthcare), or Atto647N were attached via a 5-C 6 -aminoallyl-deoxythymidine.” Isn't that a T base? Am I confused or could the authors clarify this point?

Response: We apologise for this error. We normally work with dyes on T, and this description was not updated for the present work. We have clarified the dyes are linked to either T or C bases in the Methods.

Comment 2) Methodological detail – The surface passivation and immobilization scheme for the TIRF experiments is not given. Do the authors use the PEG/PEG-biotin surface, BSA-biotin, neutravidin directly coated on a glass slide, or something else?

Response:

This information is in the cited article [*J Am Chem Soc* **137**, 16020-16023 (2015); ref. 42 in revised manuscript]. We coat the glass surfaces with PEG and PEG-biotin, followed by the addition of neutravidin and then the biotinylated DNA. We have added a sentence to the Methods to highlight this important detail.

Comment 3) My biggest concern relates to the dramatically different behavior of the Cy5 labeled (CTG)₁₀ (N=10) slipout 3WJ. It is fascinating that this one sample was completely

stabilized in the P4 state and that changing to Atto647N restored behavior in line with all the other samples. The authors' explanation is "Since the Cy5 dye is near the loop in P4, and since Cy dyes are known to stack on DNA (ref. 31,32) we attribute this to a specific dye interaction for Cy5 that is absent when Atto647N is the acceptor." This explanation seems too simple. Looking at the diagrams in Supplementary figure 5 (translating CAG to CTG), it appears that the relative position of the dye and the loop is the same in the P3 state for N=8 3WJ, in the P2 state for N=6 3WJ and in the P1 state for N=4 3WJ as it is in the P4 state for the N=10 3WJ. These other 3 states for the other CTG 3WJs are not stabilized by such positioning (Supplementary Fig. 7 and 9). Why does the Cy5 not similarly stabilize these other states as it does for the P4 state for N=10? There must be a more complex explanation than the one given by the authors in the text.

Regardless of the detailed explanation, the large difference in dynamics for (CTG)₁₀ (N=10) 3WJ labeled with Cy5 compared to Atto647N confirms that the dye can impact the transitions among states in these 3WJs. This effect reduces the overall confidence in the quantitative kinetics reported for all the samples in the manuscript. The confidence in the kinetic parameters could be increased by some other control experiments, for example observing that the kinetics for some other 3WJs are consistent when labeling with Cy5 compared to labeling with Atto647N, or by observing consistent results in a single 3WJ that is labeled with Cy5 at different positions. Without such studies, there is little known about the influence of dye-DNA interaction in other experiments and the accuracy of the kinetic results is questionable.

I commend the authors for identifying this issue with Cy5 in the (CTG)₁₀ 3WJ, finding a solution by changing to Atto647N, and reporting the issue in the paper. This effect of the dye impacting the balance of states of these 3WJs does not reduce the publishability of the study, but I do suggest it warrants more extensive discussion in the text and also reduces overall confidence in the quantitative analysis of the kinetics.

Response:

Every mobile 3WJ showed rich kinetics except for the original (CTG)₁₀ slipout. We (and many others) have shown that Cy dyes have a propensity for specific (and non-specific) stacking on DNA, RNA and proteins, so we suspected that this was a dye artefact. We exchanged the Cy5 for Atto647N and recovered the rich dynamics and broad FRET histograms, which was good evidence that the Cy5 was only stacking in that particular sequence.

To further support this we have now designed a new mobile (CTG)₁₀ 3WJ, with exactly the same sequence but in which the point of attachment of Cy5 is shifted by three nucleotides. This prevents the Cy5 from being close to the hairpin loop in any of the possible positional isomers. This new data is now included as Supplementary Figure 22. As can be seen, this sample shows multiple FRET states and complex dynamics.

We recently showed that simply moving Cy3 by a few nucleotides from the end of ssDNA was enough to prevent stacking in a nick or gap [Stacking-induced fluorescence increase reveals allosteric interactions through DNA, *Nucleic Acids. Res* , 2018, 46, 11618-11626]. We are confident that the anomaly with the original (CTG)₁₀ structure is simply due to site-specific stacking of the Cy5. Since we do not focus on the absolute rates of migration, any other minor dye effects do not compromise the central conclusion of this paper, which is that branch migration occurs for these important structures.

Minor Comments

A few other minor points the authors may consider are:

a) The authors may want to be carefully examine their language at several points in the text. For example, in the first sentence of the discussion, they say “DNA secondary structure formation underlies all proposed mechanisms for triplet repeat diseases, ...” I agree that it underlies all proposed mechanisms for DNA triplet repeat expansion, but there are various models of how the diseases arise. Some do not strongly implicate the trinucleotide structure directly in the disease mechanism generating the pathology killing cells. For example, a Cynthia McMurray model of Huntington’s [PMID: 30930170] suggests that the induction of double strand breaks due to metabolic processes modified by mismatch repair activity in response to triplet repeat presence is the actual cause of the pathology in Huntington’s disease. I am not insisting this particular paper be cited, but rather just suggesting caution in the precise language used.

Response: We thank the reviewer for this comment, which is an important distinction. We have altered the sentence mentioned in the discussion and another in the introduction to clarify the role of the secondary structures.

b) In addition to the issue of dye-DNA interactions altering the energy balance of the states (point 3 above), I am not completely satisfied with how the kinetics studies are done in the paper. No real conclusions depend on the precise kinetics results, so this is just a comment for the authors as they consider possible revisions. The HMM program used to extract kinetics from the TIRF traces seems to not do a great job. From the few time traces in the paper, I see the HMM analysis finding a lot of transitions that could be noise fluctuations (For example Fig 2d, Extended fig 3a, extended fig 8c, etc.) because HMM indicates transitions where there is not anti-correlation in the donor and acceptor intensities. I know HMM is a standard program used widely in the field, but it does not seem to be doing a particularly good job with this data. For that reason, the absolute values of the kinetic parameters extracted do not seem highly confident, but the authors don’t make major conclusions based on these kinetics, so maybe it is not such a big deal.

Response: We have used HMM extensively and found it to be very reliable for kinetic analysis (e.g. of DNA hairpins). In the system studied here we have rather complex pathways with many possible FRET states, some of which are not uniquely defined. This makes a full kinetic analysis difficult. However, as the reviewer states, we are not basing our conclusions on such an analysis or on absolute rates. The key result of this study is that we observe interconversion of 3WJs, which we attribute to branchpoint migration.

c) related - the transition density plots for the cases without clear transitions seem non-intuitive to me (For example, Figure 3 n=2 case or extended figure 7a and e). What is a transition density plot without obvious transitions in the time traces?

Response: We added these to illustrate the absence of traces, but we agree with the reviewer that this is more likely to be confusing. Therefore we have removed these TDP plots from the relevant figures.

d) related - the selected traces seem to show inconsistent kinetics. This may just be a result of the specific few traces selected. For example, in figure 1 for the non-migratory 3WJ, S2 has very infrequent transitions whereas S3 has more frequent transitions, but looking at the dwell time plots in Figure 4d, and the kinetic rates in the tables, the differences in S2 and S3 look almost negligible.

Response: The time-traces are only ever a small snapshot of the dynamics and for such complex kinetic schemes, there will be a concomitant increase in trace-to-trace variation. In other words, there are no “representative traces” for all but the simplest of kinetic schemes (e.g. two-state dynamics). This is why we have ensured that we also include FRET histograms for each sample.

e) related - Intriguingly, in figure 2d, that trace seems to have some memory effect. From 0 to 20 seconds, it is mostly high FRET with transitions frequently to low FRET, whereas after 20 seconds, it is very low FRET with only a few transitions to higher FRET. The kinetics seem extremely rich in this system and are not captured well in the analysis. Regardless, very few conclusions depend on the kinetics, so maybe this is not a significant criticism.

Response:

We have assumed Markovian kinetics throughout. The trace in question was stitched together from two different traces, which is the way the HMM software works best. We usually avoid using such stitched traces to avoid such apparent discontinuities. We have replaced this panel with a continuous time section. Nevertheless, the argument in response to point d) above holds: often time traces will appear disjointed and “irregular”, simply because they are stochastic single-molecule trajectories.

f) The loss of the mid FRET populations in zero magnesium chloride (extended figure 6 a and e) made me wonder what the TIRF trace of some of these molecules would look like in the absence of magnesium. Not a major issue, but I would have welcomed inclusion of magnesium-free intensity time traces as additional examples.

Response: The mid-FRET peaks actually broaden rather than disappear. We have added TIRF time trace data for 0 mM MgCl₂ to Fig. 17 (panels k and l). It is now clearer that the traces and histograms are very similar to those recorded at higher concentrations of Mg.

g) Why do the authors choose CTG slipouts for the static 3WJ experiment and CAG slipouts for the mobile 3WJ experiments for the main figures in the paper? It might not matter, but it would provide more rigor in mapping the FRET states between the experiments instead of putting the mobile CTG slipouts in the supplement or alternatively use CAG static slipouts in the main text. It is probably ok, but given the difference in Cy5 labeled (CAG)₁₀ and (CTG)₁₀ mobile 3WJs, there might be surprises.

Response: We have now measured the corresponding static 3WJs with a (CAG)₁₀ slipout. We agree with the reviewer that this is an important control.

Supplementary Figures 1, 3, 4, 6 and 7 show structures, MFD data and TIRF data for static 3WJs with a (CAG)₁₀ slipout. The results are very similar to the CTG sample with the

exception that now all 4 positions have two FRET states, whereas for the CTG sample two FRET states could not be distinguished.

In light of these new data, we realise that the kinetic scheme illustrated in Fig. 4b needs to be more general. We have modified it to show that there are 8 possible states and that we are only reliably able to fit up to 6 FRET states, either because there are only 6 unique FRET values or due to the limited resolution of the method. However, this resolution is more than enough to prove that we have exchange of positional isomers.

h) Possible typo – in captions of extended figure 2e and extended figure 3e, do they mean TIRF FRET instead of MFD FRET?

Response: These typos have been corrected.

i) Possible typo– in captions for Supplementary figure 2d, do they mean S4? S3 is repeated twice in the caption.

Response: This typo has been corrected.

j) Occasionally the color scheme is not consistent within the figures of the paper. For example, in supplementary figure 4, sometimes purple is the highest FRET state, but sometimes it is not.

Response: We have corrected the colour scheme throughout.

REVIEWER #2

We thank this reviewer for their careful review and helpful comments, which we have addressed in detail below.

Comment 1) For SM-FRET measurements, it is inevitable of labeling the target, oligonucleotides, in these studies. For the fully complementary duplex, I assume that there would be no interaction of fluorophore and nucleotide bases, but in the hairpin region of CAG trinucleotide repeats, there is an Adenine-Adenine mismatched base pair next to the Cy5-labeled cytosine. I would like to see the comments from the authors if there were any possibilities of interaction between the Cy5 chromophore and the A-A mismatch. If yes, how the interaction affected the bulge migration through the Cy5 labeled cytosine.

Response: As discussed in the response to reviewer 1 (Comment 3), the only clear discrepancy that we observed was with the mobile (CTG)₁₀ 3WJ. In general, the results were very similar for all structures between CAG and CTG structures, so we do not believe that there are strong interactions with the A-A mismatches. However, as we also noted above, we cannot rule out a small influence on the absolute rates. The key result from this manuscript is that interconversion/migration occurs for these structures. The similar behaviour found for different lengths of slipouts, two different repeats (CAG vs CTG) and using two different single-molecule methods gives confidence that this is a general result.

Comment 2) In figure 3, the authors investigated the dependence of dynamics on repeat length by changing the repeat number from 2 ($n = 2$) to 30 ($n = 30$) with being the position of the labeling cytosine fixed for four CAG repeat away from the end of the repeat. (leaving 3 CAG repeat for hybridization with 3 CTG repeats). Thus, once the branchpoint migration passes the labeled cytosine, the distance between two fluorophores would likely not be much different regardless of the repeat length. While six FRET states were observed for the 3WJs with CAG repeat longer than 6, but plots showed different transition intensities for those showing six FRET stage. I do not see the discussion on these observations.

Response: The answer to this is partly covered by the response to the Minor point b) of Reviewer 1. Ideally, we would have 8 unique FRET states and be able to measure all of the state-to-state transitions. Due to the intrinsic difficulty in resolving multiple FRET states and the fact that many dye-dye distances could be the same for different structures, it was not possible here to give a full kinetic analysis. Therefore, we have focused on the things we can be sure about, which is the interconversion of conformation and branchpoint isomers.

Comment 3) In the last two paragraphs in the discussion, the authors discussed the relevancy of their observations and the possible repair processes. It is interesting, but there is a crucial difference between the possible slip-out structures formed in trinucleotide repeats in the genome and their experimental models, and that is the presence of slip-out in the opposite strand. In the expanded CAG/CTG repeat, both CAG and CTG strands may form the CAG- and CTG-hairpin structures (slip-out) after dissociation of CAG/CTG duplex for the replication and transcription. The dynamic behavior of the CAG slip-out is likely heavily affected by the dynamic behavior of the CTG slip-out in the opposite strand. In the current experimental model, there is no fluctuation conceivable in the CTG strand. Therefore, their discussion on the relevancy between their observations and the repair process would be much carefully discussed.

Response: We completely agree that it will be very interesting to see how slipouts in the opposite strand (or even slipouts further along on the same strand) can influence each other. However this is really an extension of the present work. To understand more complex structures, we first need to understand the behaviour of the 3WJs we report here. Furthermore, our structures are biologically relevant in their own right. Formation of a slipout in only one strand followed by reannealing produces the structures we have studied. Likewise, slipouts in opposite strands that are very distant from each other along the duplex are essentially a combination of the isolated CAG and CTG 3WJs that we have studied. We have now shown that such slipouts can migrate. Therefore, it will be very interesting to see if they can migrate apart or together and over what distances and timescales. This is outside the scope of the present work and may not even be suited to the single-molecule methods described here. But this work will hopefully encourage others to explore this and the implications for disease progression.

Comment 4) The recent publication (ref. 10) by Nakamori et al. described the CAG-binding small molecules induced the repeat contraction in vivo. It would be particularly interesting how CAG-binding small molecule affects the slip-out dynamics.

Response: We agree that this would be an exciting direction for further research, and we are optimistic that the RED community will see the potential in using single-molecule techniques to complement existing methods.

REVIEWERS' COMMENTS

Reviewer #1 (Remarks to the Author):

The revisions of the text and the inclusion of the new control experiments have addressed my previous comments. The responses in the rebuttal to the other points in the previous review are satisfactory. I have no other suggestions for the manuscript. I congratulate the authors on a nice paper.

Reviewer #2 (Remarks to the Author):

I have looked at the responses from authors and found these responses are fully satisfactory to me.

Thus, I have no further concern to this manuscript.